# The effect of NASM-based corrective exercises on lumbar lordosis angle and selected muscle activity in women with lower cross syndrome: A randomized clinical trial

**Somayyeh Ghaffari**◉**, Seyed Mohammad Hosseini**◉*, Mehdi Gheitasi**

Department of Health and Rehabilitation in Sports, Faculty of Sport Sciences and Health, Shahid Beheshti University, Tehran, Iran.

* moh_hosseini@sbu.ac.ir

## Abstract

### Background

Lower Cross Syndrome (LCS) is a complex musculoskeletal condition characterized by muscle weakness and tightness patterns, typically resulting from prolonged and repetitive activities or inactivity.

### Objective

This study examined the effects of eight weeks of NASM corrective exercises on improving the lumbar lordosis angle and muscle activity in women with LCS.

### Method

This randomized controlled trial employed a pre- and post-test design. Thirty women with LCS were randomly assigned to an exercise group (n = 15) or a control group (n = 15). The exercise group underwent the NASM exercise protocol. The lumbar lordosis angle was measured using a 30 cm flexible ruler (KEARING brand, China), and muscle activity (gluteus maximus, hamstrings, erector spinae) was assessed via Myon 320 surface electromyography system (Switzerland) during maximum voluntary isometric contraction. The exercise group completed 24 sessions of the NASM protocol over eight weeks. Data were analyzed using paired t-tests and ANCOVA.

### Results

The results showed that eight weeks of NASM corrective exercises led to a significant between-group difference in the lumbar lordosis angle (F = 24.82, p = 0.001) and the electrical activity of the gluteus maximus muscle (F = 5.11, p = 0.032). Within the exercise group, significant improvements were also observed in the electrical activity of the hamstrings, the maximum voluntary isometric contraction (MVIC) of the gluteus

**Data availability statement:** All relevant data are within the manuscript and its Supporting Information files.

**Funding:** The author(s) received no specific funding for this work.

**Competing interests:** The authors have declared that no competing interests exist.

maximus and erector spinae, and the onset timing of the erector spinae activity ($p \leq 0.05$). However, in the between-group comparison, no significant difference was observed in the electrical activity of the hamstrings and erector spinae muscles, the maximum voluntary isometric contraction (MVIC), and the start time of muscle activity ($p > 0.05$ for most comparisons).

## Conclusion

The NASM corrective exercise program was effective in reducing lumbar lordosis and enhancing gluteus maximus activity in women with LCS. These findings support the use of this protocol as a specific intervention for improving these parameters in the studied population.

## Trial registration

IRCT20240805062660N1, Iranian Registry of Clinical Trials (IRCT), (March 1 to May 31, 2024).

## 1. Introduction

Lower Cross Syndrome (LCS) is a complex musculoskeletal condition characterized by muscle weakness and tightness patterns, typically resulting from prolonged and repetitive activities or inactivity [1]. This condition is prevalent among women and is often worsened by lifestyle factors such as prolonged sitting and wearing high-heeled shoes, which can lead to muscle imbalances and postural changes [2]. In this syndrome, the trunk muscles, including the rectus abdominis, internal and external obliques, and transverse abdominis, as well as the gluteal muscles (maximus, medius, and minimus), become weakened. Consequently, their functions are often compensated for by more superficial muscle groups. Simultaneously, excessive activation and stiffness are commonly observed in the spinal extensors, such as the erector spinae, multifidus, quadratus lumborum, and latissimus dorsi— as well as in the hip flexors, including the iliopsoas and sartorius muscles [3]. This imbalance typically leads to stiffness in the hip flexors and lumbar extensors, resulting in anterior pelvic tilt (APT) and an increased lumbar lordosis [4].

Lumbar lordosis is the natural inward curve of the lumbar spine. A normal lordotic curve typically ranges from 30 to 40 degrees [5]. However, excessive curvature, known as hyperlordosis, can arise from muscle imbalances. This condition is often associated with anterior pelvic rotation and a protruding abdomen [6]. This condition is not merely a cosmetic concern; it can also result in significant musculoskeletal dysfunction, particularly when associated with LCS-related imbalances [5,6]. Changes in the natural curve of the spine are influenced by various factors, with muscle imbalance being a key contributor to abnormalities in the lumbar-pelvic region. Dysfunction in the muscles that support the spine leads to increased stress on the vertebrae, changes in the spinal curvature, and subsequent lower back pain [7]. During physical activities, individuals with LCS often exhibit delayed activation of

the gluteus maximus— an essential muscle for hip extension and pelvic stabilization [8,9]. This delay can reach up to 370 milliseconds after the initiation of movement by other muscles, indicating an inefficient neuromuscular recruitment pattern. Such delays are linked to decreased walking efficiency and altered movement patterns across the kinetic chain. When lower body muscles such as the gluteus maximus are weak or inhibited, compensatory patterns may develop, affecting the function of other muscles such as the hamstrings during both exercises and daily tasks [8].

Over time, various methods have been used to correct these abnormalities, typically involving separate strength and stretching exercises. However, advancements in exercise science have resulted in new protocols specifically designed to target LCS. The National Academy of Sports Medicine (NASM) in the United States has developed a comprehensive corrective exercise protocol consisting of four phases: inhibition, lengthening (stretching), activation, and integration [10,11]. Research has demonstrated the effectiveness of this approach [10,12]. For example, Ghadirian Marnani et al. (2024) reported that an NASM corrective exercise program significantly reduced lumbar lordosis angles among participants compared to control groups [12]. Similarly, Okhli et al. (2019) confirmed these results, showing that both NASM and pilates exercises reduce lumbar lordosis, with NASM exercises yielding a greater effect [10]. Other studies have shown increased gluteus maximus activity and reduced overactivation of lumbar extensors following NASM-based interventions [12]. Samadi and Hajilo (2024) highlighted the importance of these neuromuscular changes in alleviating symptoms associated with LCS and enhancing lumbar stability and strength [13]. Electromyography (EMG) findings further support these improvements post-intervention [12].

While the effect of NASM exercises on lumbar lordosis is becoming established, the literature lacks comprehensive evidence regarding their impact on the electromyographic activity of the entire posterior kinetic chain in individuals with LCS. In particular, the response of the hamstrings to this specific intervention remains relatively unexplored, despite their crucial role in pelvic dynamics and their known involvement in compensatory patterns within LCS. A review of the existing literature highlights the importance of selecting targeted exercise interventions to address the muscle imbalances characteristic of this condition. Therefore, the present study aims to provide a more comprehensive investigation by examining the effect of an eight-week NASM corrective exercises program on lumbar lordosis angle and selected muscle activity in women with LCS, thereby specifically addressing the gap in understanding the neuromuscular adaptations of the hamstrings and other key stabilizers to this protocol.

## 2. Research methodology

The present study was a randomized controlled trial classified as applied research, registered under the clinical trial code IRCT20240805062660N1 (https://irct.behdasht.gov.ir/search/result?query=IRCT20240805062660N1). The research employed a two-group design to evaluate the effects of NASM corrective exercises on the lumbar lordosis angle and the electrical activity of selected muscles, including the gluteus maximus, hamstrings, and erector spinae, through pre-test and post-test assessments. The experimental group participated in the NASM corrective exercise program, while the control group received no intervention.

### 2.1. Population and sampling

The statistical population consisted of inactive female students aged 18–30 diagnosed with LCS during the academic year 2024–2025 in Tehran. Participants were identified through a screening process conducted in physical education classes at Allameh Tabataba'i University. Participants were recruited between March 1, 2024, and May 31, 2024. All participants provided written informed consent before participation. No minors were included in the study. Based on previous research [14] and using G*Power software version 3.1, the minimum required sample size was computed as 24, assuming a 95% confidence level, an effect size of 0.5 (representing a medium effect, based on similar studies on LCS [12,14]), and a test power of 0.8. To account for potential attrition, 30 participants were recruited using purposive sampling. Using a computer-based random number generator, an independent researcher who was not involved in recruitment or assessment

allocated participants to either the experimental or control group. The allocation sequence was concealed until groups were assigned. Participants were randomly allocated to the experimental group (n = 15) or the control group (n = 15), ensuring equal distribution across both groups (Fig 1). The inclusion criteria were as follows: female gender; age between 18 and 30 years; a diagnosis of LCS, defined as a lumbar curve angle of 45 degrees or greater and anterior pelvic tilt of 15 degrees or greater [15], and no history of spinal or lower limb surgery. The exclusion criteria included: sustaining any physical injury during the study, inability to complete the exercise protocol or undergo assessments, missing more than two training sessions, voluntarily withdrawing from the study for any reason, and regular use of tobacco, alcohol, or any medication that could affect neuromuscular function or pain perception.

## 2.2. Tools

**2.2.1 A researcher-developed questionnaire** was used to collect demographic data, including age, weight, height, injury history, and both physical and neurological health status.

**2.2.2 Height and weight measurement.** Height was measured using a digital stadiometer (Seca model 206) with an accuracy of 0.1 cm, and weight was measured using a digital scale (Seca model 767) with an accuracy of 0.1 kg; both devices were manufactured in Germany.

**2.2.3 Lumbar lordosis angle measurement.** The lumbar lordosis angle was measured using a non-invasive 30 cm flexible ruler (KEARING brand, manufactured in China) based on Yoda's method. Two anatomical landmarks —the twelfth

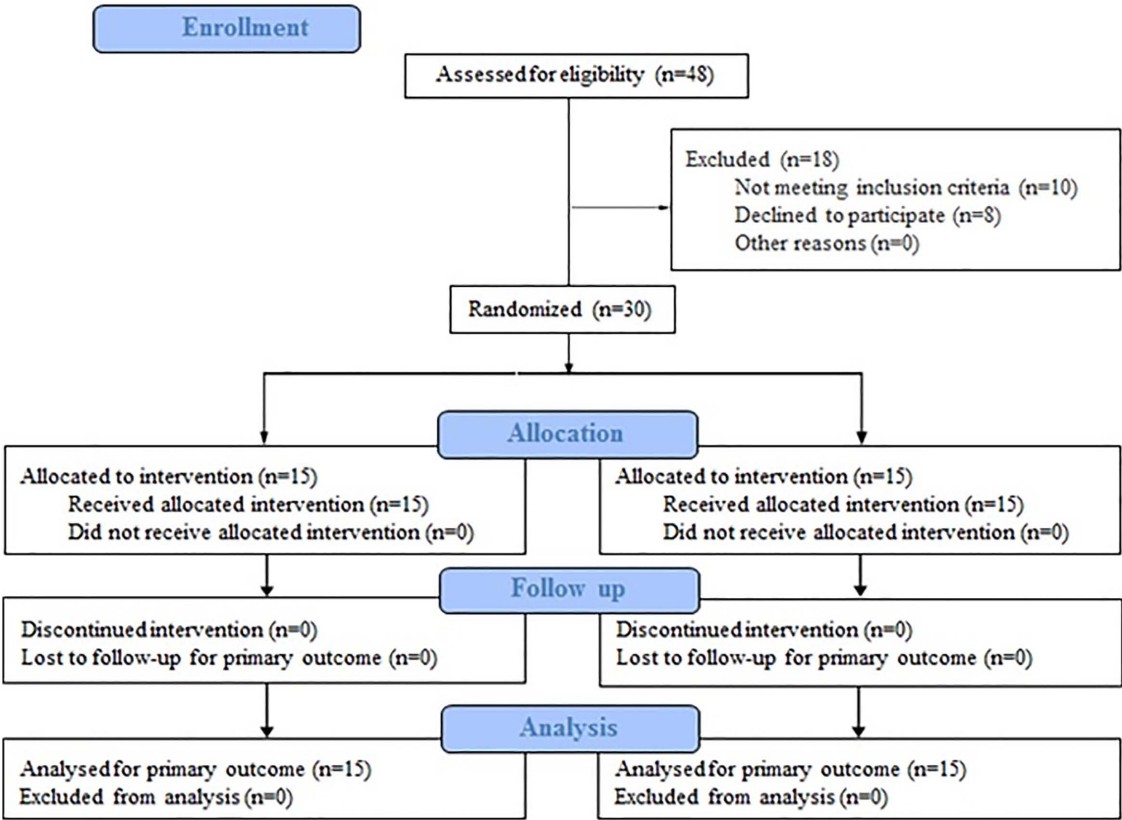

**Fig 1. Diagram of the progress through the phases of a randomized trial of two groups (that is, enrolment, intervention allocation, follow-up, and data analysis).**

thoracic vertebra (T12) and the second sacral vertebra (S2)—were identified and marked. The flexible ruler was aligned along the natural curve of the lumbar spine, and the contact points were marked on the skin using a marker. The curve was then traced onto a white sheet of paper. The angle was calculated using the formula $\theta = 4[\arctan(2H/L)]$, where L is the arch's chord length of the curve, and H is its height (with the T12-S2 line considered as the vertical reference). This formula yields the lumbar lordosis angle in degrees. Each participant underwent three separate measurements. To ensure consistency, all measurements were performed by the same experienced researcher, using a method with high reported reliability (ICC = 0.97) [13].

**2.2.4 Maximum voluntary isometric contraction (MVIC) of selected muscles.** Surface electromyography (EMG) was performed using a Myon 320 system (16-channel, wireless; Myon AG, Switzerland). Electrode placement followed the standardized SENIAM protocol to ensure consistency and reproducibility. Bipolar disposable surface electrodes (SKIN-TACT, F-55 model; 2 cm diameter, 2 cm inter-electrode distance; silver/silver-chloride material; Leonhard Lang, Austria) were applied. The raw EMG signals were sampled at 2000 Hz. Signal processing was performed using MATLAB software (MathWorks, USA). The raw data were first band-pass filtered (20–500 Hz), and a 50 Hz notch filter was applied to remove mains electricity interference. The MVIC task involved hip hyperextension in a prone position. Initially, participants were instructed to remain relaxed without muscle contraction for a few seconds to establish baseline EMG activity. Once the EMG signal reached a predefined threshold, it was considered the onset of muscle activation [16–18]. Participants were asked to hold each contraction for 5 seconds, with three repetitions conducted. One minute of rest was provided between trials to minimize the risk of muscle fatigue. During each trial, EMG signals were recorded for the gluteus maximus, hamstrings, and erector spinae (Fig 2) (S1 Fig). The maximum muscle activity was defined as the highest root mean square (RMS) value observed across the MVIC trials and was used to normalize subsequent muscle activity data. For the analysis, the RMS value during the 2nd to 5th second of each trial was calculated. The average RMS value during this interval was then divided by the maximum RMS value to obtain normalized muscle activity, expressed as a percentage of MVIC. This normalization enabled more accurate comparison of muscle activity across different testing conditions [19].

## 2.3. Method of execution

This study was designed as a randomized controlled trial with pre-test and post-test assessments. The independent variable was the NASM corrective exercise program, while the dependent variables included the lumbar lordosis angle and the activity of selected muscles. Pre-test measurements were conducted to evaluate the lumbar lordosis angle and the timing and intensity of muscle activity. Following the initial assessments, the intervention group participated in a structured exercise protocol consisting of 24 sessions, each lasting 60 minutes, focused on NASM corrective exercises. Upon

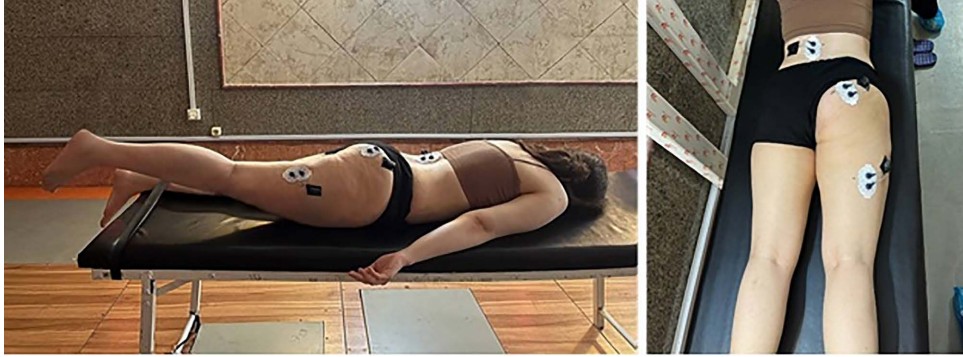

**Fig 2. Electrode placement technique for surface EMG recording of the gluteus maximus, hamstrings, and erector spinae muscles during the prone hip hyperextension task (from top to bottom).**

completion of the intervention, post-test measurements were performed for both the experimental and control groups to assess changes in lumbar lordosis and muscle activity. Before participation, all subjects provided informed consent and were explicitly informed of their right to withdraw from the study at any time, without penalty or consequences. Participant has provided consent for publication: The individual shown in Figs 2 and 3 and in the Supporting Information files has provided written informed consent (as outlined in the PLOS consent form) to publish their image alongside the article. The control group was notified that if they found the exercise protocol's outcomes satisfactory, they would have the opportunity to participate in the intervention after the study's completion. The study strictly adhered to ethical guidelines and regulations for human research. The study protocol was approved by the Ethics Committee of the Physical Education and Sports Sciences Research Institute (ethical code: IR.SSRC.REC.1402.233). All assessments and intervention sessions were conducted at the Corrective Movement and Pathology Laboratory at Shahid Beheshti University, Tehran, as scheduled.

### 2.4. Intervention

The NASM exercise protocol consisted of four phases: inhibition, stretching, activation, and integration. Participants performed the exercise program following a five-minute standard warm-up [20], as illustrated in Fig 3. In Weeks 1 and 2, the protocol began with inhibition exercises, performed for one set of 30 seconds. In Weeks 3 and 4, the duration of inhibition exercises increased to one set of 45 seconds, becoming the primary focus of the sessions, while stretching exercises were introduced (one set of 45 seconds). During Weeks 5 and 6, inhibition and stretching exercises continued with an increased volume of three sets of 60 seconds as required, alongside the introduction of activation exercises (12 repetitions per set), which became the main component of the training. Finally, in Weeks 7 and 8, integration exercises (15 repetitions per set) were added to the routine, along with continued inhibition, stretching (one set of 60 seconds), and activation (15 repetitions per set) exercises [21].

### 2.5. Statistical analysis

In this study, descriptive and inferential statistics were utilized for data analysis. The normality of the data was assessed using the Shapiro-Wilk test. To ensure the groups were comparable at baseline, independent samples t-tests were conducted on all pre-test outcome variables. To compare the pre-test and post-test results within each group, paired t-tests were conducted. For evaluating the differences between the experimental and control groups while controlling for baseline measures, multivariate analysis of covariance (ANCOVA) was applied. A significance level of $p \leq 0.05$ was considered for statistical analyses. All analyses were performed using SPSS software (version 26).

### 3. Results

The independent t-test results for participants' demographic characteristics and all baseline outcome variables are presented in Table 1. The groups were homogeneous in terms of age, weight, and height ($p > 0.05$). However, baseline comparisons revealed significant differences between the groups in the pre-test values of EA Hamstrings ($p = 0.003$), MVIC Hamstrings ($p = 0.029$), and MVIC Erector Spinae ($p = 0.017$). These baseline variables were therefore included as covariates in their respective ANCOVA models to control for pre-existing differences.

To evaluate within-group changes from pre-test to post-test, paired t-tests were conducted, and the findings are presented in Table 2. The results indicated that eight weeks of NASM exercises led to significant improvements in the exercise group across multiple variables, including lumbar lordosis angle, electrical activity of the hamstrings, and MVIC of the gluteus maximus and erector spinae muscles.

ANCOVA was performed to compare the groups at the post-test stage while controlling for relevant baseline measures, with results shown in Table 3. The analysis revealed a significant difference between the control and exercise groups in terms of lumbar lordosis angle (F = 24.82, p = 0.001) and the electrical activity of the gluteus maximus muscle (F = 5.11,

week 1 and 2 training schedule | week 3 and 4 training schedule

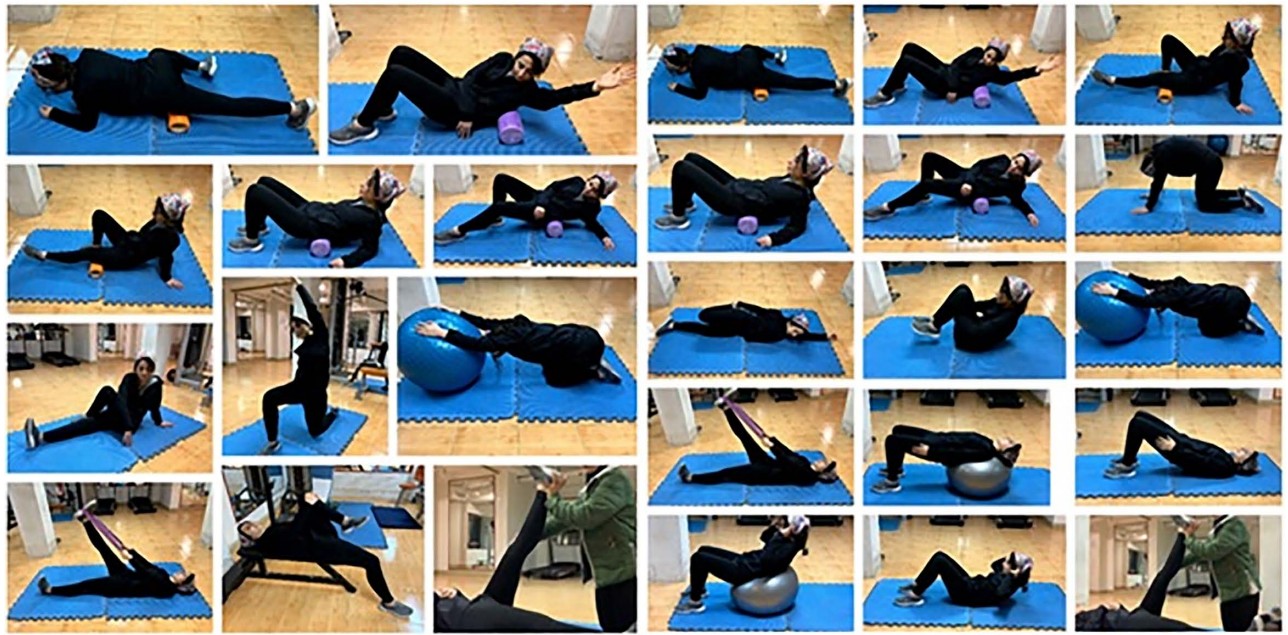

week 5 and 6 training schedule | week 7 and 8 training schedule

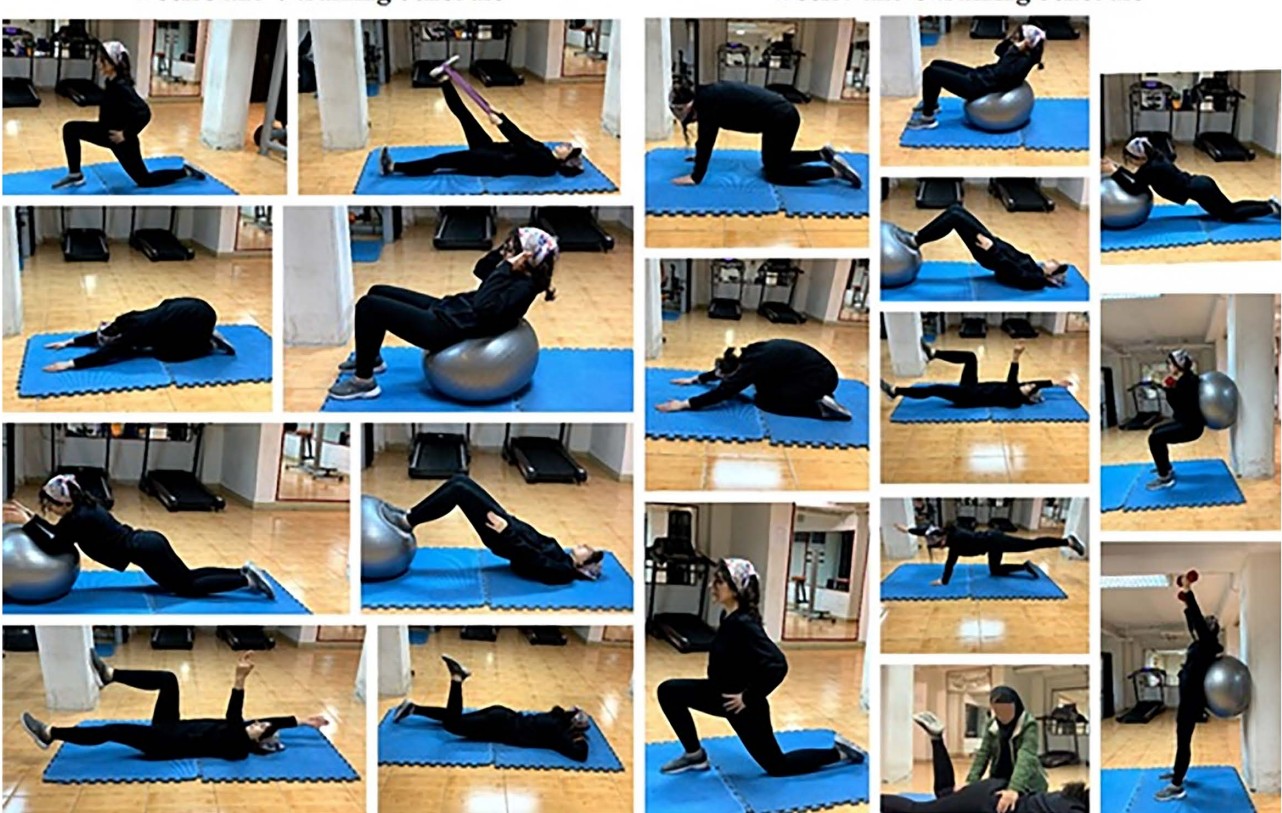

**Fig 3. NASM training protocol.**

**Table 1. Baseline Characteristics and Clinical Outcomes of the Study Participants.**

| Variable | Control Group | Exercise Group | p-value |
|---|---|---|---|
| Age (years) | 24.40±2.2 | 24.21±3.1 | 0.855 |
| Weight (kg) | 61.52±6.6 | 59.05±6.1 | 0.313 |
| Height (cm) | 165.4±5.4 | 161.8±5.2 | 0.083 |
| Lumbar Lordosis (deg) | 49.78±9.94 | 49.99±8.06 | 0.771 |
| EA Hamstrings (%MVIC) | 3.50±1.73 | 5.31±1.17 | 0.003* |
| EA Gluteus Maximus (%MVIC) | 5.15±3.10 | 7.64±3.66 | 0.059 |
| EA Erector Spinae (%MVIC) | 5.40±3.58 | 7.23±4.58 | 0.242 |
| MVIC Hamstrings (mV) | 0.26±0.10 | 0.18±0.06 | 0.029* |
| MVIC Gluteus Maximus (mV) | 0.12±0.06 | 0.11±0.04 | 0.564 |
| MVIC Erector Spinae (mV) | 0.29±0.18 | 0.16±0.06 | 0.017* |
| Start Time of EA Hamstrings (s) | 0.13±0.22 | 0.01±0.28 | 0.226 |
| Start Time of EA Gluteus Maximus (s) | −0.15±0.27 | −0.19±0.27 | 0.695 |
| Start Time of EA Erector Spinae (s) | 0.04±0.23 | 0.09±0.38 | 0.642 |

Mean (standard deviation); EA: Electrical Activity (%MVIC); MVIC: Maximum Voluntary Isometric Contraction (mV); Significance level was set at p≤0.05.

**Table 2. Within-group Pre- and Post-test Comparisons of Lumbar Lordosis Angle and Muscle Activity.**

| Variable | Control Group (n=15) | | | | Exercise Group (n=15) | | | |
|---|---|---|---|---|---|---|---|---|
| | Pre-test | Post-test | t | p | Pre-test | Post-test | t | p |
| Lumbar Lordosis (deg) | 49.78±9.94 | 51.58±10.01 | −0.901 | 0.384 | 49.99±8.06 | 44.03±6.67 | 4.828 | 0.000* |
| EA Hamstrings (%MVIC) | 3.50±1.73 | 4.37±8.76 | −2.617 | 0.021* | 5.31±1.17 | 9.99±7.29 | −2.192 | 0.046* |
| EA Gluteus Maximus (%MVIC) | 5.15±3.10 | 5.36±2.96 | −0.422 | 0.679 | 7.64±3.66 | 13.19±11.38 | −1.724 | 0.107 |
| EA Erector Spinae (%MVIC) | 5.40±3.58 | 9.86±9.22 | −1.764 | 0.101 | 7.23±4.58 | 9.09±5.69 | −0.908 | 0.379 |
| MVIC Hamstrings (mV) | 0.26±0.10 | 0.18±0.13 | 3.290 | 0.066 | 0.18±0.06 | 0.14±0.09 | 1.052 | 0.310 |
| MVIC Gluteus Maximus (mV) | 0.12±0.06 | 0.17±0.74 | −3.223 | 0.007* | 0.11±0.04 | 0.14±0.06 | −3.551 | 0.003* |
| MVIC Erector Spinae (mV) | 0.29±0.18 | 0.24±0.17 | −0.031 | 0.975 | 0.16±0.06 | 0.23±0.09 | −2.484 | 0.026* |
| Start Time of EA Hamstrings (s) | 0.13±0.22 | 0.12±0.33 | −3.346 | 0.055 | 0.01±0.28 | −0.16±0.23 | 1.935 | 0.073 |
| Start Time of EA Gluteus Maximus (s) | −0.15±0.27 | −0.33±0.27 | −1.152 | 0.152 | −0.19±0.27 | −0.37±0.29 | 1.994 | 0.066 |
| Start Time of EA Erector Spinae (s) | 0.04±0.23 | −0.04±0.35 | −0.840 | 0.416 | 0.09±0.38 | −0.14±0.21 | 2.391 | 0.031* |

Mean (standard deviation); EA: Electrical Activity (%MVIC); MVIC: Maximum Voluntary Isometric Contraction (mV); Significance level was set at p≤0.05.

p=0.032), after controlling for baseline measures. Partial eta-squared values were calculated as the effect size, with interpretations from Kesselman et al. (1998) as follows: 0.01 for a small effect, 0.06 for a moderate effect, and 0.14 for a large effect [22].

## 4. Discussion

The primary aim of this randomized controlled trial was to evaluate the impact of an eight-week NASM corrective exercise program on lumbar lordosis and muscle activity in women with LCS. A key methodological consideration was the presence of significant baseline differences in hamstrings electrical activity and strength parameters between groups, despite random allocation. This finding underscores the value of our rigorous analytical approach, which uses ANCOVA to control for these pre-existing differences, thereby providing a more accurate estimate of the true intervention effect [22].

 

**Table 3. Between-group Comparison of Adjusted Post-Test Outcomes Using ANCOVA.**

| Variable | Control Group | Exercise Group | Degrees of Freedom | F-value | p-value | Effect Size (Eta Squared) |
|---|---|---|---|---|---|---|
| Lumbar Lordosis (deg) | 51.13 | 44.45 | 1 | 24.82 | 0.001* | 0.418 |
| EA Hamstrings (%MVIC) | 8.83 | 10.47 | 1 | 0.21 | 0.650 | 0.008 |
| EA Gluteus Maximus (%MVIC) | 5.40 | 13.15 | 1 | 5.11 | 0.032* | 0.164 |
| EA Erector Spinae (%MVIC) | 9.83 | 9.11 | 1 | 0.060 | 0.808 | 0.002 |
| MVIC Hamstrings (mV) | 0.16 | 0.17 | 1 | 0.023 | 0.881 | 0.001 |
| MVIC Gluteus Maximus (mV) | 0.17 | 0.18 | 1 | 0.288 | 0.596 | 0.011 |
| MVIC Erector Spinae (mV) | 0.24 | 0.28 | 1 | 1.075 | 0.309 | 0.040 |
| Start Time of EA Hamstrings (s) | −0.14 | −0.15 | 1 | 0.000 | 0.934 | 0.007 |
| Start Time of EA Gluteus Maximus (s) | −0.26 | −0.37 | 1 | 0.943 | 0.340 | 0.035 |
| Start Time of EA Erector Spinae (s) | −0.14 | −0.15 | 1 | 1.137 | 0.296 | 0.042 |

Mean (standard deviation); EA: Electrical Activity (%MVIC); MVIC: Maximum Voluntary Isometric Contraction (mV); Start Time of EA (seconds); Significance level was set at $p \leq 0.05$.

After controlling for baseline values, our results demonstrate that the NASM protocol produced significant between-group improvements in its primary targets: lumbar lordosis angle and gluteus maximus electrical activity. The reduction in lumbar lordosis ($F = 24.82$, $p = 0.001$) aligns with previous research employing the NASM approach [12,13]. For instance, Ghadirian Marnani et al. (2024) reported a significant decrease in lumbar lordosis following an 8-week NASM intervention, a finding that is directly comparable to our results [12]. Similarly, Okhli et al. (2019) demonstrated that both NASM and Pilates exercises reduced lumbar lordosis in female students, with NASM exercises yielding a superior corrective effect, further validating the efficacy of this specific protocol [10]. The consistency of our findings with these studies strengthens the evidence base for using NASM exercises to improve spinal alignment. This effect can be attributed to the protocol's systematic focus on inhibiting and lengthening shortened hip flexors and lumbar extensors and simultaneously activating and integrating inhibited gluteal and abdominal muscles [23]. This rebalancing likely promotes a more neutral pelvic position, reducing the mechanical stress on the lumbar vertebrae and intervertebral discs [21,24].

This finding is mechanistically sound, as the NASM training approach specifically targets the muscle imbalances inherent to LCS by strengthening weakened muscles (such as the glutes and abdominals) and stretching tight muscles (including the hip flexors and spinal erectors) to promote a more neutral pelvic and spinal alignment [23]. The lordosis angle is a critical parameter for spinal biomechanics [25], and its excessive increase can lead to detrimental structural and functional changes [26]. Therefore, NASM exercises—designed to enhance stabilizing muscle function and neuromuscular control—can effectively contribute to the reduction of aberrant spinal curvatures [27]. The decrease in lordosis can be attributed to the exercises' role in strengthening the deep abdominal and lumbar muscles, thereby creating greater stability in the lumbar region and reducing mechanical stress on the vertebrae and intervertebral discs [21,24].

A novel and clinically significant finding was the robust between-group improvement in gluteus maximus electrical activity ($F = 5.11$, $p = 0.032$). This demonstrates the protocol's efficacy in addressing a fundamental impairment in LCS—gluteal inhibition. Our results are in partial agreement with Samadi and Hajilo (2024), who also observed improvements in muscle function following NASM exercises [13]. However, our study provides more precise EMG evidence of enhanced neuromuscular recruitment of the gluteus maximus, a crucial stabilizer for hip extension and pelvic control [28,29]. This objective EMG evidence addresses a gap in the literature regarding the specific neuromuscular adaptations induced by NASM exercises in this population.

The interpretation of results for other muscles requires careful consideration. While within-group improvements were observed for hamstrings electrical activity and MVIC of the erector spinae in the exercise group, the lack of significant between-group differences after controlling for baseline suggests the protocol's effect was most specific to its core

objectives. The observed within-group improvement in hamstring activity in our study contrasts with the findings of Lehman et al. (2004), who reported a dominant role of the hamstrings as compensators for gluteal weakness during hip extension [8]. This discrepancy could be attributed to methodological differences; our protocol specifically targeted gluteal activation, which may have reduced the compensatory demand on the hamstrings, leading to a less pronounced change in their overall activity in the between-group comparison. Furthermore, the unexpected significant change in hamstring electrical activity within the control group may be attributed to measurement variability, unaccounted daily activities, or habituation to testing procedures, highlighting the importance of using a controlled design to isolate the specific effect of the intervention. To achieve comprehensive neuromuscular adaptations across the entire posterior chain, future protocols might consider incorporating more targeted exercises for the hamstrings and erector spinae.

Regarding muscle activation timing, the within-group improvement in erector spinae onset ($p = 0.031$) suggests a trend toward enhanced motor control. This finding partially supports the work of Kim et al. (2014), who investigated muscle onset times in individuals with hyperlordosis [16]. While between-group differences in timing parameters were not significant, the observed patterns indicate potential benefits that warrant further investigation in larger trials with longer intervention periods.

This study has several strengths and limitations. Among its strengths are the randomized controlled design, the use of objective outcome measures (EMG), and the adherence to a standardized, well-defined intervention protocol. Limitations include the relatively small sample size of young women, which may limit the generalizability of the findings to other populations, such as men or older adults. The absence of a long-term follow-up assessment also means that the durability of the observed effects remains unknown. Furthermore, the use of a flexible ruler, while non-invasive and reliable, is not the gold-standard method for measuring spinal curvature compared to radiography. Future studies should investigate the long-term effects of NASM exercises, include a more diverse population, and explore the effects of supplementing the protocol with targeted exercises for the hamstrings and erector spinae to elicit more comprehensive neuromuscular adaptations.

## 5. Conclusion

The findings of this randomized controlled trial demonstrate that an eight-week NASM corrective exercise program is an effective intervention for reducing lumbar lordosis angle and improving gluteus maximus electrical activity in women with Lower Cross Syndrome. These results provide objective EMG evidence supporting the neuromuscular efficacy of the NASM approach for this population. The primary improvements were observed in the core targets of the protocol—spinal alignment and gluteal muscle function. While some within-group improvements were noted in other muscles, the lack of significant between-group differences for the hamstrings and erector spinae suggests the protocol's effect is most pronounced on its primary corrective targets. A key limitation of this study is its focus on a specific group of young women, which may affect the generalizability of the findings. Future research should investigate the long-term sustainability of these benefits and explore the effects of this protocol in broader populations, including men and older adults. Based on our results, the NASM corrective exercise protocol can be confidently recommended as a specific and effective non-pharmacological approach for managing lumbar hyperlordosis and gluteal muscle inhibition in women with LCS.

## Supporting information

**S1 Fig. Representative raw EMG signals.** Raw electromyography signals from the gluteus maximus, hamstrings, and erector spinae muscles during the MVIC task pre- and post-intervention.
(JPG)

**S1 File. CONSORT_2025_Checklist_Filled.**
(DOCX)

**S2 File. Original_Protocol_English.**
(DOCX)

## Acknowledgments

We would like to express our sincere gratitude to all the participants and individuals who contributed to and cooperated in this study. Their valuable involvement and support were essential to the success of this research.

## Author contributions

**Conceptualization:** Seyed Mohammad Hosseini, Mehdi Gheitasi.

**Data curation:** Somayyeh Ghaffari, Seyed Mohammad Hosseini.

**Formal analysis:** Somayyeh Ghaffari.

**Investigation:** Somayyeh Ghaffari.

**Methodology:** Somayyeh Ghaffari.

**Project administration:** Seyed Mohammad Hosseini.

**Software:** Somayyeh Ghaffari, Mehdi Gheitasi.

**Supervision:** Seyed Mohammad Hosseini.

**Validation:** Mehdi Gheitasi.

**Visualization:** Seyed Mohammad Hosseini.

**Writing – original draft:** Seyed Mohammad Hosseini.

**Writing – review & editing:** Seyed Mohammad Hosseini.

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
