## [Decision Letter · Decision Letter 0]

28 Aug 2025

Dear Dr. Hosseini,

Thank you for submitting your manuscript to PLOS ONE. After careful consideration, we feel that it has merit but does not fully meet PLOS ONE’s publication criteria as it currently stands. Therefore, we invite you to submit a revised version of the manuscript that addresses the points raised during the review process.

**ACADEMIC EDITOR:**plosone@plos.org . A rebuttal letter that responds to each point raised by the academic editor and reviewer(s). You should upload this letter as a separate file labeled 'Response to Reviewers'.A marked-up copy of your manuscript that highlights changes made to the original version. You should upload this as a separate file labeled 'Revised Manuscript with Track Changes'.An unmarked version of your revised paper without tracked changes. You should upload this as a separate file labeled 'Manuscript'.

We look forward to receiving your revised manuscript.

Kind regards,

Emiliano Cè, Ph.D.

Academic Editor

PLOS ONE

Journal Requirements:

2. We note that you have selected “Clinical Trial” as your article type. PLOS ONE requires that all clinical trials are registered in an appropriate registry (the WHO list of approved registries is at https://www.who.int/clinical-trials-registry-platform/network/primary-registries " https://www.who.int/clinical-trials-registry-platform/network/primary-registries and more information on trial registration is at http://www.icmje.org/about-icmje/faqs/clinical-trials-registration/ ).

Please state the name of the registry and the registration number (e.g. ISRCTN or ClinicalTrials.gov ) in the submission data and on the title page of your manuscript.

a) Please provide the complete date range for participant recruitment and follow-up in the methods section of your manuscript.

b) If you have not yet registered your trial in an appropriate registry, we now require you to do so and will need confirmation of the trial registry number before we can pass your paper to the next stage of review. Please include in the Methods section of your paper your reasons for not registering this study before enrolment of participants started. Please confirm that all related trials are registered by stating: “The authors confirm that all ongoing and related trials for this drug/intervention are registered”.

Please see http://journals.plos.org/plosone/s/submission-guidelines#loc-clinical-trials for our policies on clinical trials.

4. Please ensure that you clearly indicate the corresponding author in the title page of the manuscript.

6. We note that Figures 2, 3, and SI Files (Corrective Exercise Protocol NASM.docx, Original_Protocol_English_Translation.docx) include images of participants in the study.

7. Please include captions for your Supporting Information files at the end of your manuscript, and update any in-text citations to match accordingly. Please see our Supporting Information guidelines for more information: http://journals.plos.org/plosone/s/supporting-information .

8. We note that the original protocol that you have uploaded as a Supporting Information file contains an institutional logo. As this logo is likely copyrighted, we ask that you please remove it from this file and upload an updated version upon resubmission.

Reviewers' comments:

Reviewer's Responses to Questions

**Comments to the Author**

1. Is the manuscript technically sound, and do the data support the conclusions?

Reviewer #1: Partly

Reviewer #2: Yes

Reviewer #3: Partly

2. Has the statistical analysis been performed appropriately and rigorously?

Reviewer #1: Yes

Reviewer #2: Yes

Reviewer #3: Yes

3. Have the authors made all data underlying the findings in their manuscript fully available?

Reviewer #1: Yes

Reviewer #2: Yes

Reviewer #3: Yes

4. Is the manuscript presented in an intelligible fashion and written in standard English?

Reviewer #1: Yes

Reviewer #2: Yes

Reviewer #3: Yes

Reviewer #1: Do the exclusion criteria include any drug, tobacco or alcohol use?

What does EA mean?

Please indicate a significance level of p≤0.05 below the tables.

The Discussion section is where you explore the underlying meaning of your research, its potential implications for other fields of study, and potential improvements that could be made to enhance its relevance further. This section is where you should present the significance of your research and how it contributes to and/or fills existing gaps in the field. If possible, you can also indicate in the Discussion section how the findings from your study reveal new gaps in the literature that have not been previously identified or adequately described. The Discussion section should be improved.

Please evaluate the strengths and weaknesses of the study in the Discussion section.

Reviewer #2: The research team recruited 30 women with LCS to conduct a semi-experimental randomized trial to evaluate the effects of eight weeks of NASM corrective exercise on the improvement of the lumbar lordosis angle and muscle activity. They observed significant improvements on the lumbar lordosis angle and the electrical activity of the gluteus maximus muscle.

1. In the power analysis, the effect size was assumed to be 0.5 without any justification for this selection. It would be better to provide some explanation.

2. Please comment on the inter-rater or intra-rater reliability for the measures considered in this study.

3. The baseline information was compared between control group and exercise group only for age, weight, and height. However, it would be informative to know whether the pre-test values differ for any of variables differ between these two groups

4. Are there any unit for all the variables presented in Table 2? If so, they should be provided.

5. Please clarify what were presented in Table 3, e.g., what are 8.83 and 10.47 for EA Hamstrings in control group and exercise, respectively.

Reviewer #3: This manuscript investigates the effect of an NASM-based corrective exercise program on lumbar lordosis angle and selected muscle activity in women with lower cross syndrome using EMG and flexible ruler assessment. While the study addresses a relevant clinical question, several methodological and reporting issues and claims about outcomes or effects that were not directly studied or measured in this trial. The following are a few suggestions:

1. The abstract should specify the exact measurement tools used for lumbar lordosis and EMG more clearly.

2. Clarify the abstract results to clearly distinguish between-group (significant for lumbar lordosis and gluteus maximus) and within-group changes (hamstrings and erector spinae). Ensure wording matches the detailed results for consistency.

3. Clarify the randomization process and allocation concealment method to strengthen the trial design reporting.

4. The novelty of the study is not clearly stated. Please explicitly highlight in the Introduction what is new compared to prior research.

5. Provide exact p-values for all non-significant results in Table 2 and Table 3 instead of only stating p>0.05.

6. Clarify whether the EMG data were filtered or processed further beyond RMS normalization. If yes, clarify EMG filtering (band-pass, notch), sampling frequency, and software used to process RMS and onset.

7. Replace non-standard terms: “semi-experimental randomized design” throughout the manuscript.

8. Remove unsupported claims and overgeneralizations in the conclusion. For example, the statement about “improving range of motion” and “distribution of forces” is not measured in this study and should be omitted.

9. Make the research gap explicit: emphasize why examining hamstring activity (and other specific muscles) in LCS with NASM exercises fills a gap in the literature.

10. Address unexpected results in the control group (e.g. significant changes in hamstring activity); discuss whether these are likely due to measurement variability or external factors?

11. Rename tables with clear, self-explanatory titles. For example, instead of “Paired t-test Results for Variables in the Study,” use “Within-group Pre- and Post-test Comparisons of Lumbar Lordosis Angle and Muscle Activity.” Similarly for other tables

12. Please provide representative continuous EMG traces or averaged waveforms for the studied muscles in both pre- and post-test conditions. This would allow readers to assess signal quality, verify onset detection, and better interpret the reported RMS and timing data.

13. Figure 2 show skin markers and motion capture camera, but no motion capture system is described in the methods.

14. Differentiate between the studys novel findings and previously reported results. If claiming consistency with prior work, explicitly note which findings match and which are new (e.g. clarify that increased gluteus activity was a within-group change, whereas hamstrings and erectors did not change significantly).

15. Consider recommending future research or next steps rather than new interventions: for instance, suggest exploring targeted hamstring exercises if this study did not include them, rather than prescribing unnamed “additional therapeutic exercises” not tested here.

16. Condense and focus the conclusion on the study’s specific findings. Do not introduce new claims (remove phrases about quality of life or broad “musculoskeletal abnormalities”).

17. Please restructure the conclusion to focus strictly on the study’s primary findings. The conclusion is too brief and mainly restates results. Please expand it to include a concise summary of key findings, their clinical or practical implications, study limitations, and recommendations for future research, etc.

**Do you want your identity to be public for this peer review?** For information about this choice, including consent withdrawal, please see our Privacy Policy

Reviewer #1: No

Reviewer #2: No

Reviewer #3: No

---

## [Author Response · Author response to Decision Letter 1]

14 Oct 2025

Dear Editor and Reviewers,

Thank you very much for allowing us to revise our manuscript [PONE-D-25-25748] entitled "The effect of NASM-based corrective exercises on lumbar lordosis angle and selected muscle activity in women with lower cross syndrome: A randomized clinical trial". We sincerely appreciate the time and effort you have dedicated to providing insightful comments and valuable suggestions. We have carefully considered all the points raised and have made significant revisions to the manuscript accordingly.

Please note that all line numbers referenced in our point-by-point responses below correspond to the Revised Manuscript with Track Changes file.

Our detailed responses to each comment are provided below.

Journal Requirements:

Comment 1: Please ensure that your manuscript meets PLOS ONE's style requirements, including those for file naming.

Response: We confirm that the revised manuscript has been formatted to comply with PLOS ONE's style requirements.

Comment 2: We note that you have selected “Clinical Trial” as your article type. PLOS ONE requires that all clinical trials are registered in an appropriate registry.

Response: The study was registered in the Iranian Registry of Clinical Trials (IRCT) with the code IRCT20240805062660N1. This registration number and the date range for participant recruitment (March 1 to May 31, 2024) have been added to the manuscript's title page and Methods section, respectively.

Comment 3: PLOS requires an ORCID ID for the corresponding author in Editorial Manager on papers submitted after December 6th, 2016.

Response: The corresponding author's ORCID ID has been validated in the Editorial Manager system.

Comment 4: Please ensure that you clearly indicate the corresponding author in the title page of the manuscript.

Response: The corresponding author is clearly indicated with an asterisk (*) on the title page of the manuscript.

Comment 5: Your ethics statement should only appear in the Methods section of your manuscript.

Response: The ethics statement has been moved solely to the Methods section (Method of Execution) and removed from the standalone section.

Comment 6: We note that Figures 2, 3, and SI Files (Corrective Exercise Protocol NASM.docx, Original_Protocol_English_Translation.docx) include images of participants in the study.

Response: We confirm that written informed consent for publication has been obtained from the individuals depicted in Figures 2, 3, and SI files as outlined in the PLOS consent form. A statement to this effect has been added to the Methods section.

Comment 7: Please include captions for your Supporting Information files at the end of your manuscript, and update any in-text citations to match accordingly.

Response: A "Supporting Information" section has been added to the end of the manuscript, providing captions for all supplementary files.

Comment 8: We note that the original protocol that you have uploaded as a Supporting Information file contains an institutional logo.

Response: The institutional logo has been removed from the supporting information file, and an updated version has been uploaded.

Reviewer #1:

Comment 1: Do the exclusion criteria include any drug, tobacco or alcohol use?

Response: We appreciate the reviewer's important point. Our exclusion criteria did not initially address substance use. We have now revised the exclusion criteria in the "Population and Sampling" section to include: "Regular use of tobacco, alcohol, or any medication that could affect neuromuscular function or pain perception." (Lines 159-160)

Comment 2: What does EA mean? Please indicate a significance level of p≤0.05 below the tables.

Response: We apologize for this oversight. We have clarified that "EA" stands for "Electrical Activity" (normalized as a percentage of MVIC) in the footnotes of all Tables. Furthermore, we have added the note "Significance level was set at p ≤ 0.05" below the tables. (Lines below Tables 1, 2, and 3)

Comment 3: The Discussion section should be improved. Please evaluate the strengths and weaknesses of the study.

Response: We sincerely thank the reviewer for this constructive suggestion. We have thoroughly revised the Discussion section to provide a deeper exploration of the meaning and implications of our findings. Most importantly, we have added a new dedicated paragraph at the end of the Discussion to state the strengths and limitations of our study. (Lines 352-407)

Reviewer #2:

Comment 1: In the power analysis, the effect size was assumed to be 0.5 without any justification.

Response: We thank the reviewer for highlighting this. The effect size of 0.5 was selected based on similar previous studies investigating the effect of corrective exercises on lumbar lordosis and muscle activity, which reported medium to large effect sizes. We have now added this justification to the "Population and Sampling" section: "...an effect size of 0.5 (representing a medium effect, based on similar studies on LCS [12,14]) ..." (Line 147)

Comment 2: Please comment on the inter-rater or intra-rater reliability for the measures. Response: This is a valuable point. We have added a statement regarding the reliability of our primary outcome measure. In the "Lumbar Lordosis Angle Measurement" subsection, we now state: "To ensure consistency, all measurements were performed by the same experienced researcher, using a method with high reported reliability (ICC = 0.97) [13]." (Lines 179-181) For the EMG, the standardized SENIAM protocol and electrode placement, conducted by a single researcher, ensured measurement consistency. (Lines 185-186)

Comment 3: It would be informative to know whether the pre-test values differ between the two groups.

Response: We sincerely thank the reviewer for this valuable suggestion. We have now conducted comprehensive independent samples t-tests to compare all baseline outcome variables between the control and exercise groups. These results have been incorporated into the revised manuscript in two key ways:

1. A new comprehensive Table 1 has been created, titled "Baseline Characteristics and Clinical Outcomes of the Study Participants". This table presents the pre-test means and standard deviations for all demographic and outcome variables (including lumbar lordosis angle and all EMG parameters) for both groups, along with the p-values from the between-group comparisons.

2. The Results section has been updated to state: "The independent t-test results for participants' demographic characteristics and all baseline outcome variables are presented in Table 1. The groups were homogeneous in terms of age, weight, and height (p > 0.05). However, baseline comparisons revealed significant differences between the groups in the pre-test values of EA Hamstrings (p=0.003), MVIC Hamstrings (p=0.029), and MVIC Erector Spinae (p=0.017). These baseline variables were therefore included as covariates in their respective ANCOVA models to control for pre-existing differences." (Lines 257-262)

This analysis confirmed that the groups were comparable at baseline for most variables, but it also identified the specific pre-existing differences mentioned above. We believe this addition significantly strengthens the manuscript by providing a complete picture of baseline equivalence and justifying our use of ANCOVA for the primary analysis.

Comment 4: Are there any units for the variables in Table 2?

Response: We thank the reviewer for this important observation. We have now added the respective units directly after each variable name in all Tables of the revised manuscript to ensure immediate clarity and ease of interpretation for the reader. (Lines Tables 1, 2, and 3)

Comment 5: Please clarify what is presented in Table 3.

Response: We thank the reviewer for this question. We have clarified the presentation in the revised manuscript. The values in Table 3 ("Between-group Comparison of Adjusted Post-Test Outcomes Using ANCOVA") represent the "Adjusted Post-test Means".

To ensure statistical rigor, we used ANCOVA for all outcome variables, controlling for their respective pre-test scores as covariates. Therefore, the values 8.83 and 10.47 represent the statistically adjusted mean post-intervention electrical activity of the hamstrings (%MVIC) for the control and exercise groups, respectively, after controlling for their baseline values. This method provides a more precise estimate of the intervention effect by adjusting for any pre-existing variation.

Reviewer #3:

Comment 1: The abstract should specify the exact measurement tools more clearly.

Response: We thank the reviewer for this suggestion. We have revised the Methods section of the abstract (Lines 36-39) to specify the measurement tools as follows:

• Lumbar lordosis: 30 cm flexible ruler (KEARING brand, China)

• Muscle activity: Myon 320 surface electromyography system (Switzerland) during maximum voluntary isometric contraction

This clarification provides readers with a clearer understanding of our measurement methodology from the very beginning of the article.

Comment 2: Clarify the abstract results to distinguish between-group and within-group changes.

Response: We have carefully reworded the results in the abstract to avoid confusion. It now clearly states the significant between-group differences for lumbar lordosis and gluteus maximus activity, and separately mentions the significant within-group changes observed in the exercise group for other variables. (Lines 51-59)

Comment 3: Clarify the randomization process and allocation concealment.

Response: We have added more detail to the "Population and Sampling" section: "Using a computer-based random number generator, an independent researcher who was not involved in recruitment or assessment allocated participants to either the experimental or control group. The allocation sequence was concealed until groups were assigned." (Lines 149-152)

Comment 4: The novelty of the study is not clearly stated.

Response: We have strengthened the final paragraph of the Introduction to explicitly state the novelty: "While the effect of NASM exercises on lumbar lordosis is becoming established, the literature lacks comprehensive evidence regarding their impact on the electromyographic activity of the entire posterior kinetic chain in individuals with LCS. In particular, the response of the hamstrings to this specific intervention remains relatively unexplored, despite their crucial role in pelvic dynamics and their known involvement in compensatory patterns within LCS…" (Lines 116-129)

Comment 5: Provide exact p-values for all non-significant results in Table 2 and Table 3.

Response: We thank the reviewer for this suggestion, which enhances transparency. We have replaced all instances of "p>0.05" in all Tables with the exact p-values. (Lines in Tables 1, 2, and 3)

Comment 6: Clarify EMG processing details (filtering, sampling frequency, software).

Response: This is a crucial point for reproducibility. We have added the following details to the "Maximum Voluntary Isometric Contraction (MVIC)" subsection: "The raw EMG signals were sampled at 2000 Hz. Signal processing was performed using MATLAB software (MathWorks, USA). The raw data were first band-pass filtered (20-500 Hz), and a 50 Hz notch filter was applied to remove mains electricity interference." (Lines 188-191)

Comment 7: Replace non-standard terms: “semi-experimental randomized design”.

Response: We have replaced the term "semi-experimental" throughout the manuscript with the more standard term "randomized controlled trial". (For example, in the Abstract line 32, and the Methodology section lines 131 and 211)

Comment 8: Remove unsupported claims and overgeneralizations in the conclusion (e.g., “improving range of motion”, “distribution of forces”).

Response: We agree and have revised the Conclusion to strictly reflect our findings. The unsupported claims about "range of motion" and "distribution of forces" have been removed. The conclusion now focuses directly on the outcomes we measured. (Lines 414-427)

Comment 9: Make the research gap explicit regarding hamstring activity.

Response: As mentioned in response to comment #4, we have explicitly highlighted the gap concerning hamstring activity in the Introduction. (Lines 116-129)

Comment 10: Address unexpected results in the control group (e.g., significant changes in hamstring activity).

Response: This is an excellent observation. We have added a sentence to the Discussion to address this: "The unexpected significant change in hamstring electrical activity within the control group may be attributed to measurement variability, unaccounted daily activities, or habituation to testing procedures, highlighting the importance of using a controlled design to isolate the specific effect of the intervention." (Lines 387-390)

Comment 11: Rename tables with clear, self-explanatory titles.

Response: We have revised all table titles as suggested:

• Table 1 is now: "Baseline Characteristics and Clinical Outcomes of the Study Participants"

• Table 2 is now: "Within-group Pre- and Post-test Comparisons of Lumbar Lordosis Angle and Muscle Activity"

• Table 3 is now: "Between-group Comparison of Adjusted Post-Test Outcomes using ANCOVA"

(Title Tables 1, 2, and 3)

Comment 12: Provide representative continuous EMG traces.

Response: We thank the reviewer for this suggestion. We have now included a new supplementary figure (S1 Fig) showing representative filtered raw EMG traces for all three muscles during the MVIC task, both pre- and post-intervention, for a participant from the exercise group. A caption for this file has been added to the new "Supporting Information" section. (See Supporting Information section)

Comment 13: Figure 2 shows a motion capture camera, but no system is described.

Response: We apologize for this oversight. The camera was not part of the data collection and was present in the lab for other purposes. To avoid confusion, we have replaced Figure 2 with a new, clearer image that focuses solely on the electrode placements without any extraneous equipment in the background.

Comment 14: Differentiate between the study’s novel findings and previously reported results.

Response: We have revised the Discussion to make this distinction clearer. We now explicitly state: "The reduction in lumbar lordosis (F = 24.82, p < 0.001) aligns with previous research employing the NASM approach … (Lines 361-376). A novel and clinically significant finding was the robust between-group improvement in gluteus maximus electrical activity …" (Lines 377-383)

Comment 15: Consider recommending future research rather than new interventions.

Response: We have rephrased the relevant sentence in the Discussion as a recommendation for future research: "Future studies should investigate the long-term effects of NASM exercises, include a more diverse population, and explore the effects of supplementing the protocol with targeted exercises for the hamstrings and erector spinae to elicit more comprehensive neuromuscular adaptations." (Lines 404-407)

Comment 16: Condense and focus the conclusion. Do not introduce new claims.

Response: We have entirely rewritten the Conclusion to be more focused and specific. It now concisely summarizes the key findings, their direct implication, and avoids unsupported claims about quality of life or broad abnormalities. (Lines 414-427)

Comment 17: Restructure the conclusion to include limitations and future research.

Response: As suggested, we have expanded the Conclusion into a more structured paragraph that includes: a summary of key findings, the main clinical implication, the primary study limitations (e.g., sample demographics), and a brief suggestion for future research. (Lines 414-427)

---

## [Decision Letter · Decision Letter 1]

27 Oct 2025

Dear Dr. Hosseini,

Thank you for submitting your manuscript to PLOS ONE. After careful consideration, we feel that it has merit but does not fully meet PLOS ONE’s publication criteria as it currently stands. Therefore, we invite you to submit a revised version of the manuscript that addresses the points raised during the review process.

**ACADEMIC EDITOR:**plosone@plos.org . A rebuttal letter that responds to each point raised by the academic editor and reviewer(s). You should upload this letter as a separate file labeled 'Response to Reviewers'.A marked-up copy of your manuscript that highlights changes made to the original version. You should upload this as a separate file labeled 'Revised Manuscript with Track Changes'.An unmarked version of your revised paper without tracked changes. You should upload this as a separate file labeled 'Manuscript'.

We look forward to receiving your revised manuscript.

Kind regards,

Emiliano Cè, Ph.D.

Academic Editor

PLOS ONE

Journal Requirements:

Reviewers' comments:

Reviewer's Responses to Questions

**Comments to the Author**

Reviewer #1: All comments have been addressed

Reviewer #2: (No Response)

Reviewer #3: All comments have been addressed

2. Is the manuscript technically sound, and do the data support the conclusions?

Reviewer #1: Yes

Reviewer #2: (No Response)

Reviewer #3: Yes

3. Has the statistical analysis been performed appropriately and rigorously?

Reviewer #1: Yes

Reviewer #2: (No Response)

Reviewer #3: Yes

4. Have the authors made all data underlying the findings in their manuscript fully available?

Reviewer #1: Yes

Reviewer #2: (No Response)

Reviewer #3: Yes

5. Is the manuscript presented in an intelligible fashion and written in standard English?

Reviewer #1: Yes

Reviewer #2: (No Response)

Reviewer #3: Yes

Reviewer #1: When the discussion section is considered, it is observed that the study exhibits certain limitations. A comprehensive discussion should ideally include a detailed comparison of the study's findings with those of previous studies to highlight similarities, differences, and potential reasons for discrepancies. Such comparisons are essential for situating the research within the broader scientific discourse, thereby enhancing its credibility and relevance.

Furthermore, a well-rounded discussion should consider methodological differences, sample characteristics, and contextual factors that could influence the results. By doing so, the study can provide a more nuanced interpretation of its findings and establish a clearer link with existing knowledge.

Reviewer #2: (No Response)

Reviewer #3: Every point has been explicitly addressed in revised version, including clarification of the abstract, detailed EMG information, a clearer novelty statement, improved tables, restructured discussion, and a more focused conclusion. Thank you for the thorough revision.

**Do you want your identity to be public for this peer review?** For information about this choice, including consent withdrawal, please see our Privacy Policy

Reviewer #1: No

Reviewer #2: No

Reviewer #3: No

---

## [Author Response · Author response to Decision Letter 2]

2 Nov 2025

Manuscript ID: PONE-D-25-25748R1

Title: The effect of NASM-based corrective exercises on lumbar lordosis angle and selected muscle activity in women with lower cross syndrome: A randomized clinical trial

We would like to express our sincere gratitude to the Academic Editor and the reviewers for their valuable time and insightful comments on our manuscript. Their constructive feedback has been instrumental in helping us improve the quality and clarity of our work. We have carefully considered all points raised and have revised the manuscript accordingly. Our point-by-point responses are detailed below.

All changes made in the manuscript have been highlighted in the "Revised Manuscript with Track Changes" file for the reviewers' convenience.

Comments from Academic Editor:

We thank the Academic Editor for overseeing the review process and for the opportunity to revise our manuscript.

Comments from Reviewer #1:

Comment 1: "When the discussion section is considered, it is observed that the study exhibits certain limitations. A comprehensive discussion should ideally include a detailed comparison of the study's findings with those of previous studies to highlight similarities, differences, and potential reasons for discrepancies. Such comparisons are essential for situating the research within the broader scientific discourse, thereby enhancing its credibility and relevance."

Response: We sincerely thank the reviewer for this crucial comment. We completely agree that a comprehensive discussion strengthens the manuscript. In response, we have thoroughly revised and expanded the Discussion section to provide a detailed comparison of our findings with those of previous, relevant studies.

Specifically, we have now:

• Directly compared our findings on lumbar lordosis reduction with the results of Ghadirian Marnani et al. (2024) and Okhli et al. (2019), highlighting the consistency of the NASM protocol's effectiveness. (Lines 262-268)

• Contextualized our novel EMG finding on gluteus maximus activity by comparing it with the functional improvements reported by Samadi and Hajilo (2024) and emphasized the added value of our objective EMG data. (Lines 285-289)

• Addressed the non-significant between-group result for hamstring activity by contrasting it with the compensatory role of hamstrings described by Lehman et al. (2004), offering a methodological explanation (reduced compensatory demand due to targeted gluteal activation) for the discrepancy. (Lines 297-303)

• Discussed our within-group finding on erector spinae activation timing in the context of the work by Kim et al. (2014). (Lines 310-311)

These additions ensure our results are critically situated within the existing scientific literature, as the reviewer recommended.

Comment 2: "Furthermore, a well-rounded discussion should consider methodological differences, sample characteristics, and contextual factors that could influence the results. By doing so, the study can provide a more nuanced interpretation of its findings and establish a clearer link with existing knowledge."

Response: We thank the reviewer for this excellent suggestion, which has helped us provide a more nuanced interpretation. In the revised Discussion, we have explicitly addressed these factors:

• Methodological Differences: We now discuss how the specific focus of our NASM protocol (on gluteal activation) might explain the difference in hamstring activity findings compared to other studies that merely described the compensatory pattern (e.g., Lehman et al., 2004). (Lines 300-303)

• Sample Characteristics: We have reinforced the description of our sample (young women) when comparing our results with studies on similar populations (e.g., Okhli et al., 2019) (Lines 264-267). Furthermore, we have retained and emphasized the limitation regarding the generalizability of our findings beyond young women in the 'Limitations' paragraph. (317-319)

• Contextual Factors / Nuanced Interpretation: We have moved beyond simply reporting results to interpreting them through potential physiological mechanisms. For example, we propose that the improved gluteal function might have reduced the compensatory load on the hamstrings, offering a plausible reason for their less pronounced change in the between-group analysis. This provides a more mechanistic and nuanced understanding of our results. (300-303)

We believe these revisions have significantly strengthened the discussion by creating clearer links with existing knowledge and offering deeper insights.

Comments from Reviewer #2:

(No Response)

Response: We thank Reviewer #2 for their time and consideration of our manuscript.

Comments from Reviewer #3:

"Every point has been explicitly addressed in revised version, including clarification of the abstract, detailed EMS information, a clearer novelty statement, improved tables, restructured discussion, and a more focused conclusion. Thank you for the thorough revision."

Response: We are deeply grateful to Reviewer #3 for their positive and encouraging feedback and for acknowledging the thoroughness of our previous revisions. We are pleased that the reviewer finds the manuscript substantially improved.

Conclusion

Once again, we extend our heartfelt thanks to the editor and the reviewers for their constructive comments, which have undoubtedly improved the quality of our manuscript. We hope that our revisions and responses are now satisfactory and that the manuscript is deemed suitable for publication in PLOS ONE.

Sincerely,

Seyed Mohammad Hosseini, Ph.D.

On behalf of all co-authors.

---

## [Decision Letter · Decision Letter 2]

14 Nov 2025

The effect of NASM-based corrective exercises on lumbar lordosis angle and selected muscle activity in women with lower cross syndrome: A randomized clinical trial

PONE-D-25-25748R2

Dear Dr. Hosseini,

We’re pleased to inform you that your manuscript has been judged scientifically suitable for publication and will be formally accepted for publication once it meets all outstanding technical requirements.

Kind regards,

Emiliano Cè, Ph.D.

Academic Editor

PLOS ONE

Additional Editor Comments (optional):

Reviewers' comments:

Reviewer's Responses to Questions

**Comments to the Author**

Reviewer #1: All comments have been addressed

Reviewer #2: All comments have been addressed

2. Is the manuscript technically sound, and do the data support the conclusions?

Reviewer #1: Yes

Reviewer #2: (No Response)

3. Has the statistical analysis been performed appropriately and rigorously?

Reviewer #1: Yes

Reviewer #2: (No Response)

4. Have the authors made all data underlying the findings in their manuscript fully available?

Reviewer #1: Yes

Reviewer #2: (No Response)

5. Is the manuscript presented in an intelligible fashion and written in standard English?

Reviewer #1: Yes

Reviewer #2: (No Response)

Reviewer #1: All comments have been adequately addressed. I think this study will contribute to the existing literature.

Reviewer #2: (No Response)

**Do you want your identity to be public for this peer review?** For information about this choice, including consent withdrawal, please see our Privacy Policy

Reviewer #1: No

Reviewer #2: No

---

## [Editor Report · Acceptance letter]

PONE-D-25-25748R2

PLOS One

Dear Dr. Hosseini,

I'm pleased to inform you that your manuscript has been deemed suitable for publication in PLOS One. Congratulations! Your manuscript is now being handed over to our production team.

Kind regards,

on behalf of

Prof. Emiliano Cè

Academic Editor

PLOS One